# Integrated Analyses of m^6^A Regulator-Mediated Methylation Modification Patterns and Tumor Microenvironment Infiltration Characterization in Pan-Cancer

**DOI:** 10.3390/ijms231911182

**Published:** 2022-09-23

**Authors:** Qingkang Cao, Yuanyuan Chen

**Affiliations:** College of Science, Nanjing Agricultural University, Nanjing 210095, China

**Keywords:** pan-cancer, N6-methyladenosine, immune infiltration, tumor microenvironment, immunotherapy

## Abstract

The invasion of immune cells in the tumor microenvironment (TME) is closely related to cancer development. Studies have demonstrated that N6-methyladenosine (m^6^A) can affect the invasion of immune cells in TME as well as cancer development. We comprehensively analyzed the RNA-seq data of 16 different cancer types based on 20 m^6^A regulators and identified two distinct m^6^A modification patterns, which were closely associated with TME cell infiltration and overall patient survival. Then, we used principal component analysis (PCA) to construct m6Ascore based on the expression of m^6^A-related prognostic genes, which can successfully predict patient survival. The low-m6Ascore subtype is characterized by more immune cell infiltration, good prognosis and lower TNM stages, while the high-m6Ascore subtype is characterized by low immune infiltration, stromal activation, and poor prognosis. m6Ascore was also closely associated with immunotherapy response and was significantly higher in complete response/partial response (CR/PR) patients than in stable disease/progressive disease (SD/PD) patients in both immunotherapy cohorts. Therefore, our study indicates that m^6^A modification plays an important role in the prognosis of pan-cancer and the formation of complex TME in pan-cancer. Our research helps to improve the cognition of m^6^A modifications at pan-cancer levels and identify more effective strategies for immunotherapy.

## 1. Introduction

In 2020, there were approximately 19.3 million new cancer cases and nearly 10 million deaths worldwide, excluding melanoma. Cancer has become the leading cause of death before the age of 70 in many countries [1]. While there have been many studies on cancer, treating cancer remains difficult because of its complex pathogenic mechanism and diversified risk factors. Therefore, it is important to identify new cancer diagnoses and treatment strategies.

More than 150 RNA modifications, such as N6-methyladenosine (m^6^A), N1-methyladenosine (m^1^A), Pseudouracil (ψ), and adenosine to inosine (A-to-I) RNA editing have been identified in nature [2]. The m6A modification was first identified in hepatocellular carcinoma mRNAs [3]. m6A is the most abundant RNA modification in nature. It is not only abundant in eukaryotes such as mammals and plants, but also in prokaryotes. m^6^A methylation is the most common and dominant methylation modification in mammals [4], in which 7676 animal genes have undergone m6A modification in their mRNAs [5]. m6A modifications were mainly concentrated on consensus motif RRACH (R = A or G, H = A, C or U), and were highly enriched near the 3’UTR and stop codons [6]. In mammals, m^6^A methylation is a dynamic reversible mRNA modification co-regulated by methyltransferases (writers), demethylases (Erasers), and binding proteins (readers). Methyltransferases such as METTL3/5/16, RBM15/15B, ZC3H13, ZCCHC4, KIAA1429, and WTAP catalyze the formation of m^6^A methylation. The removal process is mediated by two demethylating enzymes, FTO and ALKBH5. Binding proteins, such as YTHDF1/2/3, IGF2BP1, HNRNPC, and FMR1, can recognize m^6^A methylation [7]. m^6^A regulators play an important role in various biological processes and are closely related to the occurrence and development of tumors [8,9]. For example, the low expression of HNRNPC in LUAD patients has a better prognosis. In most cancers, the expression of m^6^A regulators is positively correlated with stemness indices and negatively correlated with immune invasion. The expression of many m^6^A regulators is also significantly correlated with drug sensitivity [10].

TME is mainly composed of tumor cells, stromal cells, and the extracellular matrix (ECM). Tumor progression involves tumor cells, and is closely related to other components of TME, especially immune cells [11]. For example, endothelial cells can provide nutrition for tumor cells, fibroblasts can help tumor cells metastasize through blood vessels, and immune cells can eliminate tumor cells and effectively inhibit tumor development [12]. Studies have shown that TME plays an important role in tumor development and immunotherapy, and various cells in TME, including T cells, B cells, innate immune cells, and fibroblasts, can affect the immune checkpoint block (ICB) response [13]. TME subtypes obtained by TME classification of patients with pan-cancer can predict patient survival and serve as biomarkers for immunotherapy [14]. In recent years, some studies have confirmed that some m^6^A methylation regulators can affect TME and enhance or inhibit the infiltration of some immune cells. Wang et al. showed that up-regulated METTL3 expression could promote the activation and maturation of dendritic cells (DC), while down-regulated METTL3 expression would lead to impaired DC functional maturation, reduced expression of il-12, CD40, and CD80, and inhibited T cell activation [15]. The expression of METTL3 is significantly up-regulated in most human cancers such as breast cancer, lung cancer, gastric cancer, and liver cancer. High expression of METTL3 promotes cancer development and is associated with poor prognosis in these cancers [16]. In 2019, Han et al. found that inhibition of YTHDF1 expression could enhance the anti-tumor response of CD8 + T cells and improve the therapeutic effect of PD-L1 [17]. Comprehensive analysis of immune cell infiltration mediated by various m^6^A regulators demonstrated that m^6^A modification was significantly correlated with the level of immune cell infiltration in gastric cancer and played a significant role in immunotherapy and patient prognosis. Similar results were obtained in glioblastoma, hepatocarcinoma, gastric cancer, and pancreatic cancer [18,19,20,21]. Pathways can also have a dramatic impact on immune cells in the tumor microenvironment. For example, TGF-β signaling can limit T cell trafficking to TME, inhibit B cell proliferation and promote B cell death [22,23]. WNT/β-catenin pathway activation is usually closely associated with poor spontaneous T-cell infiltration [24]. The PI3K/Akt pathway regulates the survival, migration, proliferation and polarization of macrophages [25].

We comprehensively analyzed the RNA-seq data of 16 different cancer types in the TCGA database based on 20 m^6^A regulators. From this, we identified two different m^6^A modification patterns using an unsupervised clustering method for the expression of m^6^A regulatory factors in pan-cancer. These two modification patterns were closely associated with immune cell infiltration in TME and overall patient survival (OS). Next, we identified differentially expressed genes (DEGs) of the two modification patterns and used a univariate COX proportional risk model to select the DEGs that impacted survival. Then, we constructed the m6Ascore by PCA and found that it was significantly correlated with the two modification patterns. The m6Ascore could also predict the survival of pan-cancer patients and was closely related to immune cell infiltration in TME. Finally, it was verified that the m6Ascore could successfully predict the survival and prognosis of patients in METABRIC and GSE3494 cohorts, and that it was correlated with immunotherapy response in IMvigor210 and GSE78220 immunotherapeutic cohorts.

## 2. Results

### 2.1. The Landscape of m^6^A Regulators in Pan-Cancer

We extracted 20 m^6^A regulators from pan-cancer patients, including nine writers, eight readers, and two erasers. First, we compared the expression levels of 20 regulators in normal and pan-cancer patient samples (Figure 1A). There were 18 significantly differentially expressed regulators, and all of them were significantly up-regulated in cancer samples. Among them, the expression levels of HNRNPC, RBMX, and ALKBH5 were significantly higher than other regulators, and only FTO and METTL14 showed no significant difference between cancer and pan-cancer samples. The univariate Cox proportional hazards model was constructed based on the differential genes. The results demonstrated that METTL3/14/16, WTAP, RBM15, RBM15B, VIRMA, IGF2BP1/2/3, CBLL1, ZCCHC4, ELAVL1, YTHDC1, YTHDC2, HNRNPC, RBMX, FMR1, FTO, and ALKBH5 all significantly affected survival. In particular, RBM15 had the most significant effect on survival (Figure 1B). We divided patients into low-expression and high-expression groups based on median RBM15 expression. Kaplan-Meier curves were generated according to the low-expression and high-expression groups, and patients in the low-expression group had a significant survival advantage (Appendix A). This is consistent with previous studies, which found that high expression of RBM15 is significantly associated with tumor progression and poor prognosis in a variety of cancers, including LSCC, COAD, LIHC, and PAAD [26,27,28,29]. Most m^6^A regulators were significantly positively correlated, while IGF2BP1, IGF2BP2, and IGF2BP3 were weakly correlated with other regulatory factors. Only IGF2BP2 was negatively correlated with METTL14, ALKBH5, YTHDC2, and FTO (Figure 1C and Appendix A). In conclusion, m^6^A regulators are closely related to patient survival, and there is a significant difference in expression between normal and cancer samples. There is also a significant correlation between regulators, and m^6^A regulators could play an important role in the occurrence and development of cancer.

### 2.2. m^6^A Regulator-Mediated Methylation Modification Patterns in Pan-Cancer

Based on the expression of these 20 m^6^A regulators, two distinct m^6^A modification patterns (called m6Acluster 1 and m6Acluster 2) were determined (Appendix A). Prognostic analysis of the two modification patterns demonstrated that m6Acluster 2 had a significant survival advantage (Figure 2A). There were also significant differences in the expression of m^6^A regulators between the two m^6^A modification patterns. Compared to m6Acluster 2, the expression levels of ELAVL1, HNRNPC, IGF2BP2, IGF2GP3, METTL3, and RBM15B were higher in m6Acluster 1. The expression levels of RBM15, VRMA, and WTAP showed no difference between m6Acluster 1 and m6Acluster 2, and the expression levels of other regulators were higher in m6Acluster 2 (Figure 2B).

GSVA enrichment analysis was used to explore the differences in biological functions of the two modification patterns (Appendix A). m6Acluster 1 was significantly enriched in cell cycle, P53 signaling pathway, WNT signaling pathway, basal cell carcinoma, and other cancer-related pathways. m6Acluster 2 was significantly enriched in beta-alanine metabolism and circadian rhythm mammal pathways (Figure 2C). Activation of the P53 signaling pathway and WNT signaling pathway can promote the development of cancer, thus affecting patient survival. It could be that significant enrichment of the P53 signaling pathway, the WNT signaling pathway, basal cell carcinoma, and other cancer-related pathways in m6Acluster 1 lead to the poor survival of patients in m6Acluster 1.

### 2.3. Correlation between m^6^A Modification Patterns and Cell Infiltration Characteristics of TME in Pan-Cancer

To explore the correlation between m^6^A modification patterns and cell infiltration characteristics of TME, we used the ssGSEA method to evaluate the infiltration level of 28 immune cells (Appendix A). We found that most of the 28 immune cells were more infiltrated in m6Acluster 2 (Figure 3A). We also used the ESTIMATE package to calculate the immune score, stromal score, and ESTIMATE score for the samples, all of which were higher in m6Acluster 2 (Figure 3B). At the same time, the tumor purity was lower in m6Acluster 2 (Figure 3B). Therefore, the m6Acluster 2 modification pattern has a significant survival advantage.

Activation of the epithelial-mesenchymal transition (EMT) and WNT signaling pathways are enriched in m6Acluster 1. This indicates that the stroma activity of m6Acluster 1 is significantly enhanced. CD8 effector T-cell immune-related pathways are enriched in m6Acluster 2 (Figure 3C). This suggests that the m^6^A modification pattern has a significantly different immunophenotype. m6Acluster 1 lacks effective immune infiltration, activates immunosuppressive pathways such as WNT, and has a poor prognosis. It is classified as an immune-desert phenotype. m6Acluster 2 is enriched in immune cell infiltration, activates immune-related CD8 T-effector signature, and has a good prognosis. It is classified as an immune-inflamed phenotype.

To investigate the role of m^6^A-related phenotypes in immune regulation, we extracted WNT signaling pathway-related genes, immune-related genes, and immune checkpoint-related genes from the published literature and investigated the expression of these genes in different m^6^A modification patterns. Immune-related genes, including CD8A, GZMA, PRF1, and TBX2, were significantly up-regulated in m6Acluster 2 (Figure 4A). This indicates that m6Acluster 2 was the immune-activation group. Genes relevant to the WNT pathway were up-regulated in m6Acluster 1 (Figure 4B), indicating that m6Acluster 1 was a stromal-activated group, which was consistent with the results obtained by previous enrichment analysis and immune infiltration. There was no difference in the expression of immune checkpoint-related gene TIGIT between the two groups, while the other immune checkpoint-related genes were significantly different between the two m^6^A clusters, and most were significantly up-regulated in m6Acluster 2 (Figure 4C). This suggests that m^6^A modification patterns could be associated with immune checkpoint therapy. Then, Spearman correlation analysis was used to test the correlation between m^6^A regulators and immune cell infiltration (Appendix A). m^6^A regulators had a certain correlation with infiltrating immune cells, of which IGF2BP1, IGF2BP2, and IGF2BP3 were positively correlated with most immune cells. Most other regulators were negatively correlated with immune cells (Figure 4D).

### 2.4. m^6^A Gene Signature Subtypes

A total of 40 m^6^A-related DGEs between m^6^A modification patterns were obtained, after which a univariate Cox proportional hazards model was constructed based on the DGEs (Appendix A). Thirty-seven DGEs with significant prognosis were then used for KEGG enrichment analysis (Appendix A). These genes are enriched in immune or cancer-related pathways, such as the Toll-like receptor signaling pathway, choline metabolism in cancer, and complement and coagulation cascades (Appendix A). These results indicate that m^6^A modification is closely related to immune regulation. Unsupervised cluster analysis was performed based on the expression of these prognostic DGEs, and patients were divided into two m^6^A-related gene phenotypes: m^6^A gene cluster A and m^6^A gene cluster B (Appendix A). Prognostic analysis demonstrated that m^6^A gene cluster B had a significant survival advantage (Figure 5A). Meanwhile, activation of the EMT and WNT signaling pathways were enriched in m^6^A gene cluster A. This indicates that the stroma activity of the m^6^A gene cluster A was significantly enhanced. CD8 effector T-cell immune-related pathways were enriched in m^6^A gene cluster B (Figure 5B). The level of immune infiltration in m^6^A gene cluster B was significantly higher than that of m^6^A gene cluster A (Figure 5C). A comparison of the expression of m^6^A regulators in the two m^6^A gene clusters demonstrated results that were consistent with the previous m^6^A modification patterns (Figure 5D). In the two m^6^A gene clusters, patient prognosis, immune infiltration, pathway enrichment, and expression of m^6^A regulators were consistent with the results of m^6^A methylation modification patterns. Of them, patients with m^6^A gene cluster B and m6Acluster 2 had significant survival advantages, and there was abundant infiltration of immune cells in TME. Stroma activation was observed in m^6^A gene clusters A and m6Acluster 1, and effective immune infiltration was absent in TME. This indicates that m^6^A methylation modification patterns were closely associated with cancer development and TME immune cell infiltration.

### 2.5. m6Ascore and Performance Validation

To evaluate the methylation modification of a single patient in pan-cancer, we used 37 DEGs with prognostic effects to construct the m6Ascore using PCA. The R package “Maxstat” was used to calculate the optimal cut-off value, and patients were divided into the low-m6Ascore group and the high-m6Ascore group according to m6Ascore (Appendix A). Prognostic analysis showed that patients with a low m6Ascore had a significant survival advantage (Figure 6A). We found that patients with low m6Ascore had higher levels of immune cell infiltration, and CD8 effector T-cell immune-related pathways were enriched in them. Meanwhile, the EMT and WNT signaling pathways were enriched in the high-m6Ascore group (Figure 6B,C). The low-m6Ascore group also had higher immune and stromal scores (Figure 6D–F), and the corresponding low-m6Ascore group had lower tumor purity (Figure 6G). This suggests that the m6Ascore in a single patient is closely related to tumor progression and immune infiltration.

We found significant differences in the expression level of m^6^A regulators between the low-m6Ascore group and the high-m6Ascore group (Figure 7A), which was consistent with the previous m6Acluster and m^6^A gene cluster. We compared the differences in m6Ascore between two m^6^A modification patterns, while m6Acluster 2 showed lower m6Ascore (Figure 7B). We also compared the difference in m6Ascore between two m^6^A gene clusters, while m^6^A gene cluster B showed lower m6Ascore (Figure 7C). This is consistent with a better prognosis for patients with lower m6Ascore. The alluvial diagram shows that almost all samples of low-m6Ascore and m^6^A gene cluster B come from m6Acluster 2 (Figure 7D). These results also show that m^6^A methylation modification is closely related to tumor progression and immune infiltration.

To further explore the function of the m6Ascore, we compared the differences in m6Ascore of the six immune subtypes from C1 to C6 introduced in [30] and found that C3, with the best survival rate, had the lowest scores (Figure 7E). In addition, m6Ascore was significantly lower in stage I/II patients than in stage III/IV patients (Figure 7F), which indicates that the m6Ascore is also a good predictor of tumor stage. In conclusion, m^6^A modification is significantly correlated with tumor immunophenotype and patient survival in pan-cancer. The m6Ascore can distinguish the level of immune infiltration and different tumor stages and has a certain prognostic ability for patients.

Finally, we validated the m6Ascore in an external data set, calculated the m6Ascore in METABRIC and GSE3494 breast cancer datasets, and tested the effect of m6Asore on survival. We found that patients with low m6Ascore had a significant survival advantage in these two datasets (Figure 8A,B). Finally, we used two immunotherapy cohorts, IMvigor210 (anti-PD-L1 cohort) and GSE78220 (anti-PD-1 cohort), to validate the correlation between m6Ascore and immunotherapy. We found that in IMvigor210, the high-m6Ascore group had a significant survival advantage (Figure 8C), and the proportion of CR/PR patients in the high-m6Ascore group was significantly higher than in the low-m6Ascore group (Figure 8D), and m6Ascore was significantly higher in CR/PR than in SD/PD patients in both immunotherapy cohorts. (Figure 8E). This suggests a significant therapeutic advantage and better clinical response in patients with anti-PD-L1 immunotherapy in the higher m6Ascore group. In IMvigor210, the tumor mutation burden (TMB) and neoantigen burden in the high-m6Ascore group were also significantly higher than those in the low-m6Ascore group (Figure 8F,G). In the GES78220 cohort, the proportion of patients with CR/PR in the high-m6Ascore group was significantly higher than in the low-m6Ascore group (Figure 8H). In summary, our constructed m6Ascore can help predict clinical responses to immunotherapy.

## 3. Discussion

Many studies have demonstrated that m^6^A regulators play an important role in tumor progression, inflammation, immune system, and drug resistance [31,32,33]. In this study, based on the RNA-seq data of 16 cancer types in the TCGA database, we analyzed the prognostic ability of m^6^A regulators in pan-cancer and found that RBM15 was the most significant adverse survival factor among 20 m^6^A regulators. The significant role of RBM15 in malignant cancer progression has been confirmed in many studies. We then divided the patients into two distinct m^6^A modification patterns based on 20 m^6^A regulators. There was a significant difference in TME immune cell infiltration between the two modification patterns. m6Acluster 2 is characterized by immune activation, and a significant survival advantage corresponds to the immune-inflamed phenotype. m6Acluster 1 is characterized by immunosuppression, activation of the EMT and WNT pathways, and poor prognosis, which corresponds to the immune-desert phenotype. The immune-desert phenotype is associated with a lack of T cell infiltration [34]. Activation of the EMT pathway in m6Acluster 1 inhibits T cells, resulting in low infiltration of immune cells in m6Acluster 1. GSVA results showed that there were significant functional differences between different m^6^A modification patterns at the pan-cancer level. Meanwhile, due to the heterogeneity among different cancer types, there were also some functional differences among different cancer types even in the same m^6^A modification pattern. Our study confirmed that m^6^A modification could play an important role in various cancers. In the future, we will perform a detailed analysis of the m^6^A regulator-mediated methylation modification patterns and tumor microenvironment infiltration characterization in each single cancer.

Thirty-seven DEGs with a significant impact on survival were selected based on two m^6^A modification patterns. According to the 37 differential genes, two distinct m^6^A gene clusters were obtained by clustering. The results showed that m^6^A modification played an important role in tumor prognosis and immunity. Finally, the m6Ascore was constructed based on 37 m^6^A-related prognostic genes: a high m6Ascore was characterized by significant enrichment of the WNT and EMT pathways, immunosuppression, poor prognosis, and an immune-desert phenotype; a low m6Ascore was characterized by an immune-inflamed phenotype.

Subsequently, we verified the prognostic ability of the m6Ascore in two breast cancer datasets, METABRIC and GSE3494, and found that the m6Ascore could significantly differentiate patient survival. Finally, we found that the m6Ascore can predict clinical response to immunotherapy in two immunotherapy cohorts: IMvigor21 (anti-PD-L1 Cohort) and GSE78220 (anti-PD-1 Cohort). The m6Ascore was significantly higher in CR/PR patients than in PD/SD patients in both immunotherapy cohorts, and TMB was positively correlated with m6Ascore. This suggests that patients with high m6Ascore have a better clinical response to immunotherapy.

There are interactions between immune cells, which jointly affect the entire TME. Orchestrating immune responses by CD4 and CD8 T cells was unraveled by a study carried out by the group of Hans Schreiber, interrogating the cooperation of CD4 and CD8 T cell responses in a model of bystander killing of cancer. Tumor-specific CD4 T cells provide signals for CD8 T cells to destroy tumor cells and effectively inhibit tumor development. Only these two cells act in concert to produce the effect of inhibiting tumor progression. If the tumor microenvironment has a large number of CD8 T cells, the lack of tumor-specific CD4 T cells will not be able to mount an immune response and therefore will not destroy the tumor cells. Similarly, only large numbers of CD4 T cells were present, but tumor growth was not inhibited without large numbers of CD8 T cells [35]. The infiltration level of activated CD4 T cells was higher in the m6Acluster 1 group and the high m6Ascore group. However, the infiltration level of CD8 T cells was lower in m6Acluster 1 group and the high m6Ascore group, which affected the synergistic effect of the two cells and prevented them from producing effective immune responses. Therefore, activated CD4 T cells did not contribute to immunity in the m6Acluster 1 group or the high m6Ascore group.

Type 2 T helper cells usually work in concert with eosinophils and macrophages to produce immunity against tumors. In addition, some studies have shown that Type 2 T helper cells can promote the development and metastasis of cancer [36]. The infiltration level of Type 2 T helper cells was higher in m6Acluster 1 group and the high m6Ascore group. However, the infiltration levels of eosinophils and macrophages was lower in m6Acluster 1 group and high m6Ascore group, which leads to the influence of synergism and the inability to produce effective immune response. Therefore, Type 2 T helper cells do not contribute to immunity in the m6Acluster 1 group or the high m6Ascore group. 

In conclusion, a comprehensive analysis of multiple m^6^A mediator-mediated modification patterns can provide a more accurate prognosis for patients. It is helpful to understand the m^6^A regulator-mediated complex characteristics of the mediation of TME cell infiltration. The m6Ascore we constructed can distinguish the characteristics of immune infiltration in patients and successfully predict patient survival; it can also distinguish tumor stage and is significantly correlated with TMB. Finally, the m6Ascore can also predict a patient’s immunotherapy responses. Therefore, the m6Ascore can elucidate the correlation between m^6^A and complex TME and the search for better individual-specific immunotherapy strategies. We hope that these findings can be validated in additional clinical cohorts to improve their prediction accuracy.

## 4. Materials and Methods

### 4.1. Data

RNA-seq data and clinical information data of 16 cancer types were downloaded from The Cancer Genome Atlas (TCGA) database (http://tcga-data.nci.nih.gov/tcga/ (accessed on 22 March 2022)); the number of samples for each cancer is shown in Table 1. The GSE3494 dataset, which included 251 breast cancer samples, was downloaded from the Gene-Expression Omnibus (GEO) database. The RNA-seq and clinical data of 1904 breast cancer patients were downloaded from the METABRIC database, and each patient’s m6Ascore was validated. We also downloaded two immunotherapeutic cohorts to investigate the association between m6Ascore and immunotherapy outcomes. The first one was the IMvigor210 cohort, including 348 samples (advanced urothelial cancer with anti-PD-L1 antibody altezolzumab) from the R package “IMvigor210”, and the second was the GSE78220 cohort, including 27 samples (metastatic melanoma treated with anti-PD-1 antibody Pembrolizumab) from the GEO database. The flowchart of this study is shown in Figure 9.

### 4.2. Unsupervised Clustering of 20 m^6^A Regulators

We obtained 20 m^6^A regulatory factors based on previously published literature. The 20 regulators include nine writers (METTL3/14/16, WTAP, RBM15, RBM15B, VIRMA, CBLL1, and ZCCHC4), nine readers (ELAVL1, YTHDC1, YTHDC2, HNRNPC, IGF2BP1/2/3, RBMX, and FMR1) and two erasers (FTO and ALKBH5). Based on the expression of these 20 m^6^A regulators, we performed an unsupervised cluster analysis using the PAM method in the R package “ConsensusClusterPlus”, which was repeated 1000 times to identify different m^6^A modification patterns [37].

### 4.3. Gene Set Variation Analysis (GSVA)

We downloaded the “C2.cp.kegg.v7.4. symbols” gene set from MSigDB database, which contains 186 KEGG pathways. Mariathasan et al. constructed some sets of genes related to biology, including EMT markers (including EMT1, EMT2, and EMT3), antigen processing and presentation (APAP), cell cycle, nucleotide excision repair (NER), angiogenesis, CD8 T-effector signature, the Wnt pathway, the TGF-β pathway, DNA replication, and DNA damage repair (DDR) (Appendix A) [22,38,39]. These gene sets were used to explore the correlation between m^6^A modification patterns and biological pathways. We used the R package “GSVA” for GSVA enrichment analysis to analyze the differences in biological functions of different m^6^A clusters. Functional annotation of m^6^A-related genes was performed using the R package “ClusterProfiler”.

### 4.4. Estimation of TME Cell Infiltration

We used ssGSEA (single-sample gene set enrichment analysis) to quantify the infiltration levels of 28 immune cells in pan-cancer TME (Appendix A). The gene set of 28 immune cells came from Charoentong’s study, including 13 innate immune cells and 15 adaptive immune cells [40]. The R package “ESTIMATE” was further used to assess the immune score and stromal score for each sample.

### 4.5. Generation of m^6^A Gene Signature

We constructed the m6Ascore to quantify the m^6^A modification pattern of an individual pan-cancer patient. The R package “LIMMA” was used to select the DEGs between different m^6^A modification patterns (|*log2fold change*| > 1, adjusted *p* < 0.05). Then, a univariate Cox proportional hazards model was constructed based on the DEGs. The genes with *p* < 0.05 were selected as DEGs with a significant prognosis for further analysis. Finally, PCA was used to construct the m6Ascore:(1)$$m6A\mathrm{score}=\underset{i}{\sum }\left(PC1i+PC2i\right)$$
where *i* is the expression value of each m^6^A-related prognostic gene.

### 4.6. Statistical Analysis

A Kaplan-Meier survival curve was plotted using the “Survival” package and “SurvMiner” package in R/Bioconductor 3.13 (https://www.bioconductor.org/ accessed on 15 September 2022). The R package “ezcox” constructed univariate Cox proportional Hazards model, *p* < 0.05, was considered to have a significant effect on survival. Spearman’s correlation coefficient was used to analyze the correlation between m^6^A regulators and immune cell infiltration in TME. Patients were divided into high-score and low-score groups based on m6Ascore using the R package “Maxstat.” All data analysis was performed using R4.1.0 software.

## 5. Conclusions

In this study, we comprehensively analyzed 16 different cancer types in the TCGA database based on 20 m^6^A regulators to identify different m^6^A modification patterns. First, a separate prognostic analysis was performed for each regulator. Second, two different m^6^A modification patterns were determined. Finally, the individual-specific m6Ascore was constructed. This was validated in two breast cancer datasets and two immunotherapy cohorts. Studies have assessed the m6Ascore construction based on m^6^A regulators, but they have only been analyzed in a single cancer, such as glioblastoma, hepatocarcinoma, gastric cancer, or pancreatic cancer [18,19,20,21]. We performed a comprehensive analysis of m^6^A regulator-mediated methylation modification patterns and TME characterization in 16 types of cancer. The results demonstrated that different m^6^A modification patterns had significant differences in patient prognosis and immune infiltration in TME. The m6Ascore is also a good predictor of patient survival, which is helpful for immunotherapy. Our study demonstrated that m^6^A modifications could influence TME infiltration, which is closely related to tumor occurrence and development. Therefore, studies assessing m^6^A regulator-mediated methylation modification patterns and TME infiltration characterization in pan-cancer can elucidate the relationship between m^6^A modification patterns and TME infiltration characterization and identify more effective immunotherapy strategies.

## Figures and Tables

**Figure 1 ijms-23-11182-f001:**
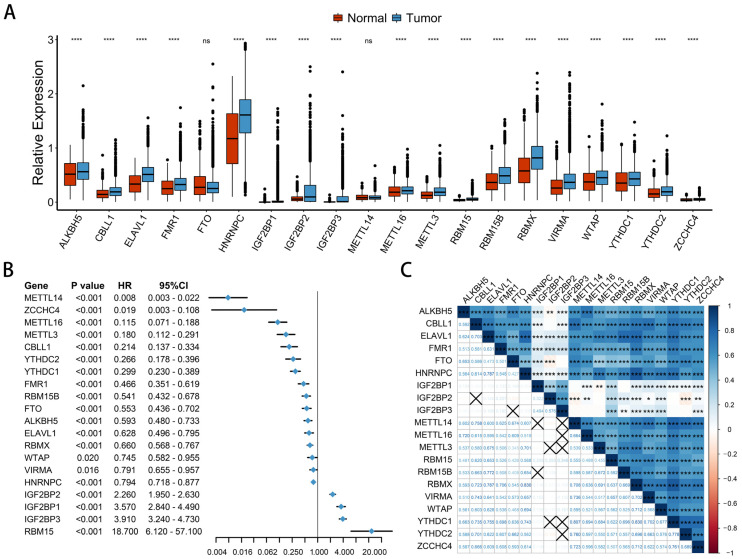
The landscape of m^6^A regulators in pan-cancer. (**A**) Expression of the 20 m^6^A regulators in normal and tumor tissues in pan-cancer. Red is normal tissue; blue is cancer tissue. The asterisks represent the statistical *p* value (**** *p* < 0.0001). (**B**) Univariate Cox regression analysis of OS in pan-cancer patients. (**C**) Relationships of the 20 m^6^A regulators in pan-cancer using Spearman analysis. Negative correlations are marked in red and positive correlations are marked in blue. The asterisks represent the statistical *p* value (* *p* < 0.05; ** *p* < 0.01; *** *p* < 0.001). The number represents the correlation coefficient, and X represents no correlation.

**Figure 2 ijms-23-11182-f002:**
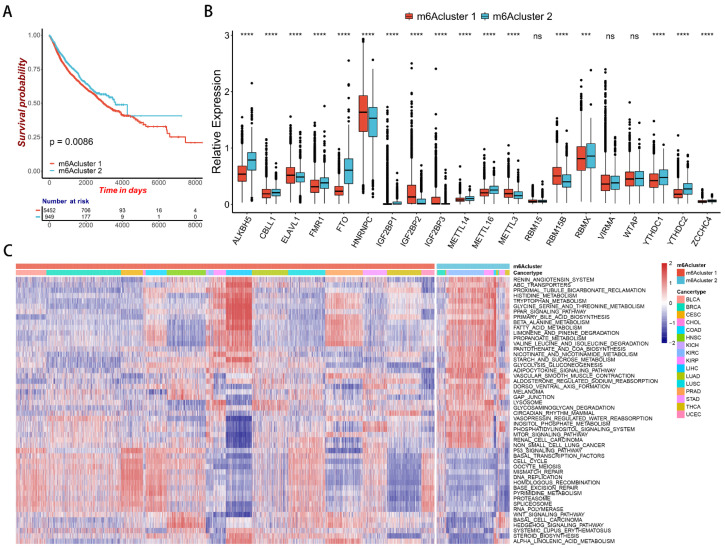
Survival and biological characteristics of m^6^A methylation modification patterns. (**A**) Kaplan-Meier curves of patients in m6Acluster 1 and m6Acluster 2 in pan-cancer. The *X*-axis shows survival in days. The *Y*-axis shows the overall survival rate. (**B**) Expression of the 20 m^6^A regulators between m6Acluster 1 and m6Acluster 2 in pan-cancer. Red is m6Acluster 1; blue is m6Acluster 2. The asterisks represent the statistical *p* value (*** *p* < 0.001; **** *p* < 0.0001). (**C**) GSVA enrichment analysis heat map. Each row represents a path; each column represents a sample.

**Figure 3 ijms-23-11182-f003:**
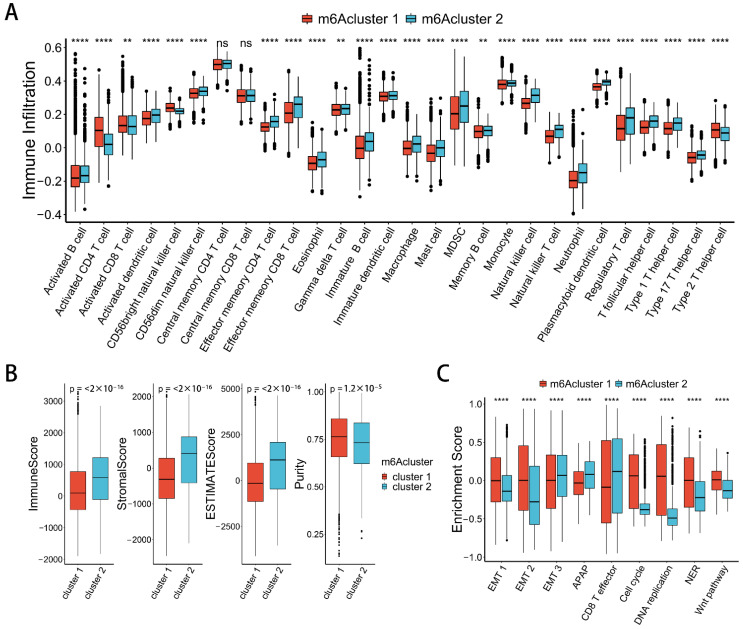
Characteristics of TME immune infiltration in two distinct m^6^A modification patterns. (**A**) Differences in the abundance of immune infiltrating cells between two m^6^A modification patterns. The asterisks represent the statistical *p* value (** *p* < 0.01; **** *p* < 0.0001). (**B**) Differences in the immune score, stromal score, estimate score, and tumor purity between two m^6^A modification patterns (Wilcoxon test). (**C**) Differences in biological functions between two m^6^A modification patterns. The asterisks represented the statistical *p* value (**** *p* < 0.0001).

**Figure 4 ijms-23-11182-f004:**
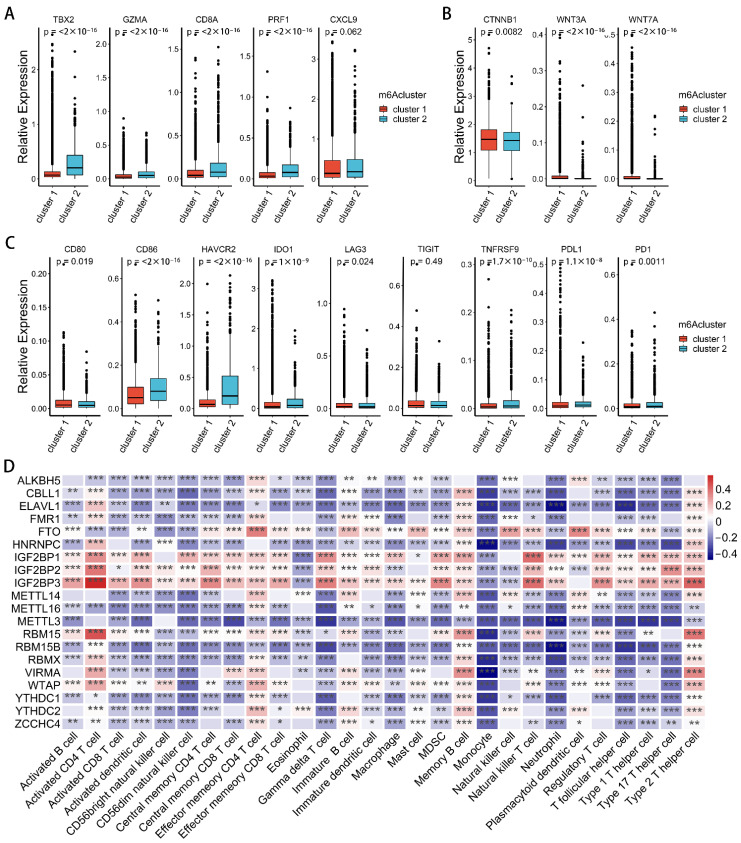
Transcriptome traits in two m^6^A clusters, and correlation between m^6^A regulators and immune infiltration in TME. (**A**) Different expressions of immune-related genes in distinct m^6^A modification patterns (Wilcoxon test). (**B**) Different expressions of WNT pathway-related genes in distinct m^6^A modification patterns (Wilcoxon test). (**C**) Different expression of checkpoint-related genes in distinct m^6^A modification patterns (Wilcoxon test). (**D**) Heat map of correlation between m^6^A regulators and levels of immune cell infiltration in TME; the *X*-axis shows 28 kinds of immune cells, and the *Y*-axis shows m^6^A regulators. Red represents positive correlations, and blue represent negative correlations. The asterisks represent the statistical *p* value (* *p* < 0.05; ** *p* < 0.01; *** *p* < 0.001).

**Figure 5 ijms-23-11182-f005:**
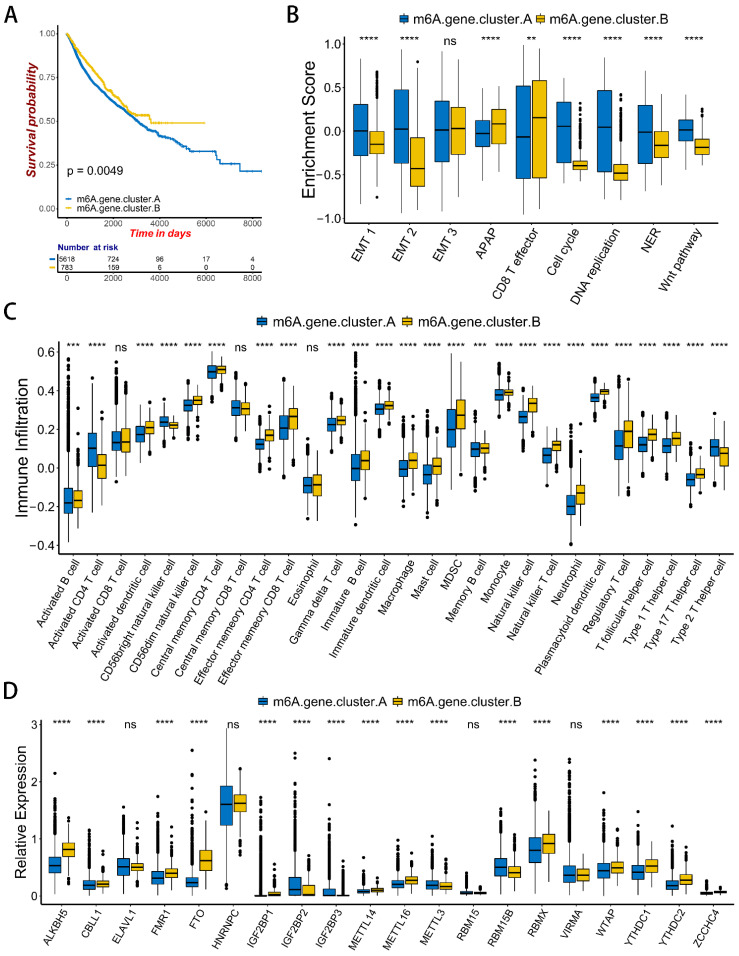
Characteristics of TME immune infiltration in two m^6^A gene clusters. (**A**) Kaplan-Meier curves of patients in m^6^A gene cluster A and m^6^A gene cluster B in pan-cancer. The *X*-axis shows survival in days. The *Y*-axis shows the overall survival rate. (**B**) Differences in biological functions between two distinct m^6^A gene clusters. Asterisks represent the statistical *p* value (** *p* < 0.01; **** *p* < 0.0001). (**C**) Difference in the abundance of immune infiltrating cells between two distinct m^6^A gene clusters. Asterisks represent the statistical *p* value (****p* < 0.001; *****p* < 0.0001). (**D**) Comparisons of the expression of 20 m^6^A regulators between m^6^A gene cluster A and m^6^A gene cluster B in pan-cancer. Blue is the m^6^A gene cluster A. Yellow is the m^6^A gene cluster B. Asterisks represent the statistical *p* value (**** *p* < 0.0001).

**Figure 6 ijms-23-11182-f006:**
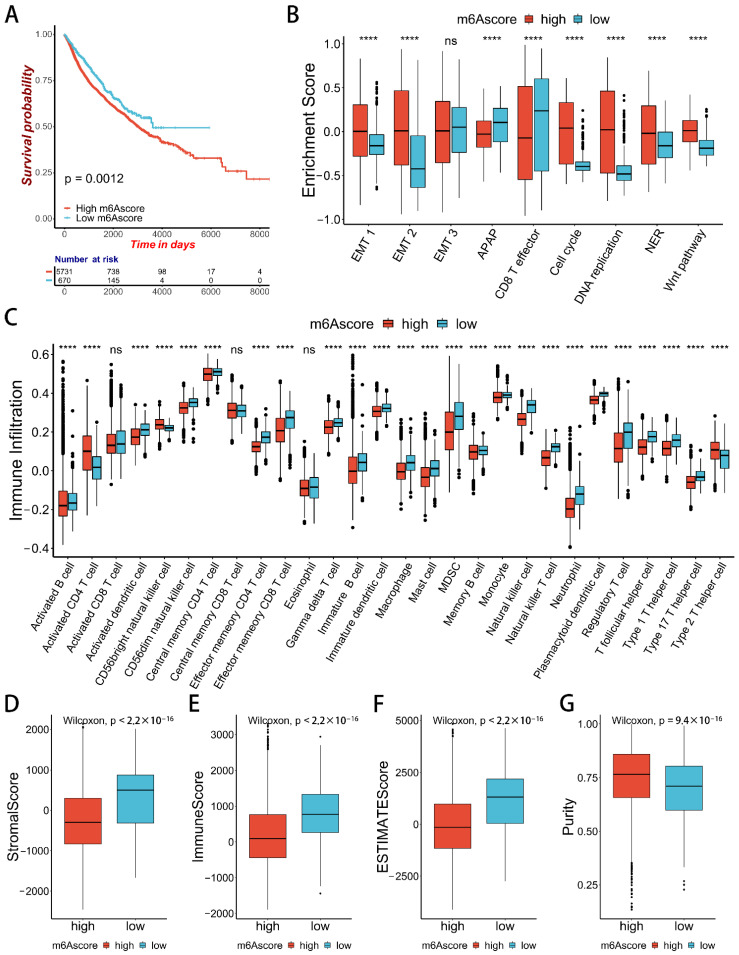
Characteristics of TME immune infiltration in two m6Ascore groups. (**A**) Kaplan-Meier curves of patients in the low-m6Ascore group and high-m6Ascore group in pan-cancer. The *X*-axis shows survival in days. The *Y*-axis shows the overall survival rate. (**B**) Differences in biological functions between two distinct m6Ascore groups. Asterisks represent the statistical *p* value (**** *p* < 0.0001). (**C**) Difference in the abundance of immune infiltrating cells between two m^6^A gene clusters. Asterisks represent the statistical *p* value (**** *p* < 0.0001). (**D**–**G**) Differences in the stromal score, immune score, estimate score, and tumor purity between the two m^6^A gene clusters (Wilcoxon test).

**Figure 7 ijms-23-11182-f007:**
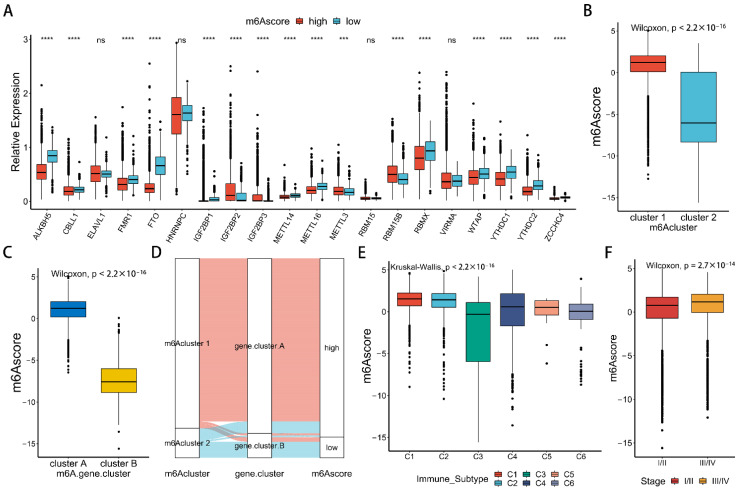
The interrelation of m6Ascores with other groups. (**A**) Comparisons of the expression of 20 m^6^A regulators between the low-m6Ascore and high-m6Ascore groups in pan-cancer. Blue is the low-m6Ascore group. Red is the high-m6Ascore group. Asterisks represent the statistical *p* value (*** *p* < 0.001; **** *p* < 0.0001). (**B**) Difference in the m6Ascore between two m6Aclusters (Wilcoxon test). (**C**) Difference in the m6Ascore between two m^6^A gene clusters (Wilcoxon test). (**D**) Alluvial diagram showing changes in m6Aclusters, m^6^A gene clusters, and m6Ascore groups. (**E**) Difference in the m6Ascore among six immune subtypes. Asterisks represent the statistical *p* value (Wilcoxon test)). (**F**) Difference in the m6Ascore between tumor stages (Wilcoxon test).

**Figure 8 ijms-23-11182-f008:**
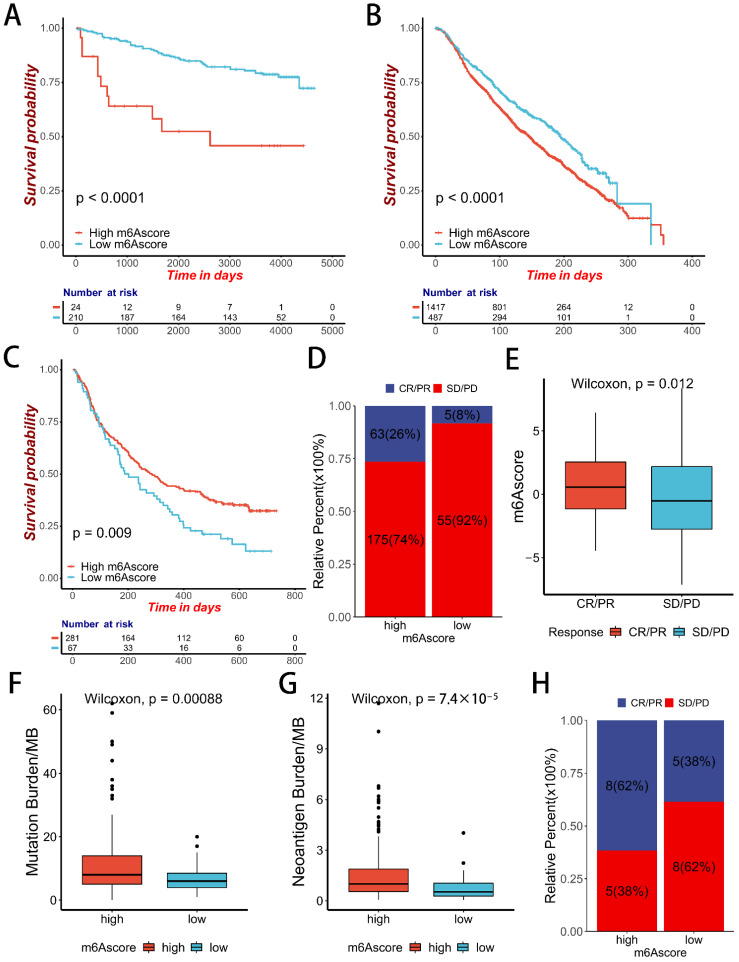
m6Ascore in the role of immunotherapy. (**A**) Kaplan-Meier curves of patients in the low-m6Ascore group and the high-m6Ascore group in GSE3494. The *X*-axis shows survival time in days. The *Y*-axis shows the overall survival rate. (**B**) Kaplan-Meier curves of patients in the low-m6Ascore group and high-m6Ascore group in METABRIC. The *X*-axis shows survival in days. The *Y*-axis shows the overall survival rate. (**C**) Kaplan-Meier curves of patients in the low-m6Ascore group and high-m6Ascore group in IMvigor210. The *X*-axis shows survival in days. The *Y*-axis shows the overall survival rate. (**D**) Proportion of IMvigor21 patients with different responses between the high-m6Ascore group and low-m6Ascore group. (**E**) Difference in the m6Ascore between two distinct response groups (Wilcoxon test). (**F**) Difference in TMB between two m6Ascore groups (Wilcoxon test). (**G**) Difference in the neoantigen burden between two distinct m6Ascore groups (Wilcoxon test). (**H**) Proportion of GES78220 patients with different responses between the high-m6Ascore group and low-m6Ascore group.

**Figure 9 ijms-23-11182-f009:**
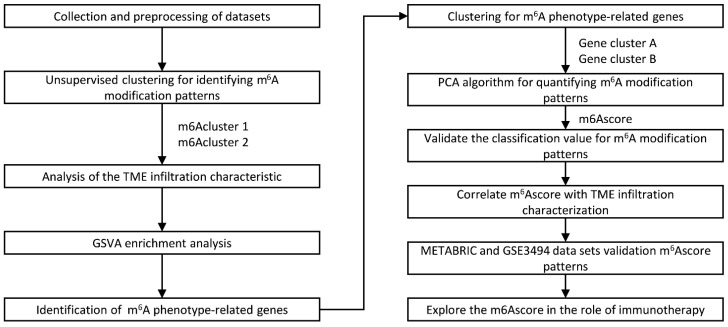
Flowchart of the approach.

**Table 1 ijms-23-11182-t001:** Number of samples of 16 types of cancer downloaded from TCGA databases.

Cancer Type	Normal Samples	Tumor Samples
Bladder Urothelial Carcinoma (BLCA)	408	19
Breast invasive carcinoma (BRCA)	1102	113
Cervical squamous cell carcinoma and endocervical adenocarcinoma (CESC)	306	3
Cholangiocarcinoma (CHOL)	36	9
Colon adenocarcinoma (COAD)	287	41
Head and Neck squamous cell carcinoma (HNSC)	522	44
Kidney Chromophobe (KICH)	66	25
Kidney renal clear cell carcinoma (KIRC)	534	72
Kidney renal papillary cell carcinoma (KIRP)	291	32
Liver hepatocellular carcinoma (LIHC)	374	50
Lung adenocarcinoma (LUAD)	517	59
Lung squamous cell carcinoma (LUSC)	502	51
Prostate adenocarcinoma (PRAD)	498	52
Stomach adenocarcinoma (STAD)	415	35
Thyroid carcinoma (THCA)	513	59
Uterine Corpus Endometrial Carcinoma (UCEC)	177	24
Total	6548	688

## Data Availability

RNA-seq data and clinical information data of 16 cancer types were downloaded from The Cancer Genome Atlas (TCGA) database (http://tcga-data.nci.nih.gov/tcga/ (accessed on 22 March 2022)). GSE3494 and GSE78220 were downloaded from the Gene-Expression Omnibus (GEO) database (https://www.ncbi.nlm.nih.gov/geo/ (accessed on 18 May 2022)). METABRIC cohort was downloaded from the METABRIC database (https://www.cbioportal.org/ (accessed on 8 June 2022)). IMvigor210 cohort was acquired from the R package “IMvigor210” (accessed on 19 May 2022). “C2.cp.kegg.v7.4. symbols” gene set was downloaded from MSigDB database (http://www.gsea-msigdb.org/gsea/msigdb/ (accessed on 10 April 2022)).

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
