# Peer review of "Integrated Analyses of m6A Regulator-Mediated Methylation Modification Patterns and Tumor Microenvironment Infiltration Characterization in Pan-Cancer"

_ijms, 2022, doi:10.3390/ijms231911182_

Round 1
Reviewer 1 Report
This is well designed study. Please include in introduction more background on the target of study (6-methyladenosine (m6A). Please, include signaling pathway Signaling to introduction or Discussion with main players in tumor microenvironment
Reviewer 2 Report
A study by Cao and Chen (Manuscript ID: ijms-1919235) is based on comprehensive analysis of 20 m6A regulators, for which data were obtained from TCGA RNA-seq database of 16 different cancer types. Authors used unsupervised clustering methods to obtain two different modification patterns and further used modification patterns to identify differentially expressed genes (DEGs) and used COX proportional risk model to determine survival probability of the DEGs. Additionally authors performed PCA analysis to construct m6A prognostic score and verified it on breast cancer cohort. Finally authors examined associations between m6A score and immunotherapy in two different immunotherapy cohorts.
I would like to congratulate the authors on this interesting study, presented in a very clear way. This is an interesting work, which is addressing an important question of m6A regulators in the complex tumor microenvironment.
Reviewer 3 Report
This manuscript described a comprehensive analysis of multiple m6A mediator-mediated modification patterns at pan-cancer levels. The authors constructed the m6Ascore to distinguish the characteristics of immune infiltration in patients and predict patient survival. They also use the m6Ascore to distinguish tumor stage and predict patients’ immunotherapy responses. The results of the study showed that m6A modification do have effects on the prognosis of pan-cancer and the formation of complex TME in pan-cancer. Although the manuscript is well prepared, there are still a few fundamental points that should be addressed:
1) In results 2.1, the authors described that METTL3 were significantly up-regulated in cancer samples, and patient in METTL3 low-expression group had a significant survival advantage. However, in line 58, they cited a study in which they showed that up-regulated METTL3 expression could promote the activation and maturation of dendritic cells (DC), while down-regulated METTL3 expression would lead to impaired DC functional maturation, reduced expression of il-12, CD40, and CD80, and inhibited T cell activation. Is this contradictory? If so, how can we use the m6Ascore to do accurate prediction? Can the authors comment on this?
2) It is interesting that, in figur3A, figure 5c, figure 6c, we can see the activated CD4 T cell and Type 2 T helper cell are both significantly decreased in the m6Acluster 2 group or m6Ascroe low group. Since these two cell types will not contribute to the immune-desert phenotype, can the authors comment on this?
3) In the introduction part, there is lots of repeated content in the last two paragraphs. Can the authors reword them?
4) Line 159, “activate” should be changed into “activates”.
5) Line 162, is there a typo in “CD8 effect T fine pathways”? Please correct it if so.
